# Diversity patterns of the South African azooxanthellate scleractinians (Cnidaria: Anthozoa), with considerations of environmental correlates

Zoleka N. Filander[1,2]*, Kerry J. Sink[3,4], Marcelo V. Kitahara[5,6], Stephen D. Cairns[6], Amanda T. Lombard[4]

1 Biodiversity and Coastal Research, Oceans and Coasts, Department of Forestry, Fisheries and Environment, Cape Town, South Africa, 2 Zoology Department, Nelson Mandela University, Gqeberha, South Africa, 3 South African National Biodiversity Institute, Cape Town, South Africa, 4 Institute for Coastal and Marine Research, Nelson Mandela University, Gqeberha, South Africa, 5 Centro de Biologia Marinha, Universidade de São Paulo, São Sebastião, Brazil, 6 Department of Invertebrate Zoology, Smithsonian Institution, Washington, DC, United States of America

* zfilander@gmail.com

**Data Availability Statement:** All relevant data are within the paper and its Supporting Information files.

## Abstract

Azooxanthellate scleractinian corals, a group of species that lack a symbiotic relationship with dinoflagellates, are influenced by environmental variables at various scales. As the global commitment to sustainably manage ocean ecosystems and resources rises, there is a growing need to describe biodiversity trends in previously unsampled areas. Benthic invertebrate research in South Africa is a developing field, and many taxa in deep water environments remain inadequately characterized. Recently, the South African azooxanthellate scleractinian fauna was taxonomically reviewed, but their distributional correlations with physical parameters have not been studied. Here we aim to understand the biodiversity gradients of the South African azooxanthellate coral fauna by analysing the environmental correlates of museum samples. The associated coordinate data were georeferenced and depth obtained from a national bathymetric dataset, prior to undertaking a multivariate analysis. This analysis encompassed several steps, including the grouping of the longitude and depth data (environmental data), identifying families characteristic of the group variability, and examining the correlation of the associated data with the biological data. Additionally, the analysis involved quantifying diversity patterns along the environmental gradients. Overall, our results confirmed two longitudinal groups (eastern margin [Group A] vs southern and western margin [Group B]) and 11 depth categories represented within two bathymetric zones (shallow [50–200 m] and deep [300–1000 m]). Caryophylliids, flabellids, and dendrophylliids contributed the most towards distinguishing longitudinal and depth gradients. Both abiotic variable (longitudinal and depth) partially explained coral distribution patterns, with depth being highly correlated to the species variation observed. Data limitations within our data set resulted to unexplained variance, however, despite these limitations, the study demonstrates that historical museum samples provide a valuable data source that can fill

**Funding:** The financial support for this research study was provided by the Department of Forestry, Fisheries, and Environment (DFFE). The funders had no role in study design, and analysis, decision to publish, or preparation of the manuscript. Some of the samples were acquired through DFFE research surveys led by lead author.

**Competing interests:** The authors have declared that no competing interests exist.

research sampling gaps and help improve the understanding of biodiversity patterns of the coral fauna in under sampled marine ecosystems.

## 1. Introduction

The distribution of azooxanthellate corals, a group of scleractinian species that lack a symbiotic relationship with photosynthetic dinoflagellates, is influenced by environmental variables at various scales [1–5]. Physical and chemical oceanographic factors, as well as geomorphologic settings affect food supply and, consequently, benthopelagic coupling [3]. Overall, depth might be used as a variable linked to several oceanographic factors that influence species distributions. For example, coral species have preferred thermal ranges [4], and a global azooxanthellate coral richness trend has been documented between 200 and 1000 m. This depth range often coincides with shelf and slope features, which may provide suitable substrate for larval settlement and habitats for azooxanthellate coral species to colonise [3, 6]. Furthermore, long-term environmental stability appears to be important for the occurrence/distribution of deep water stony coral species. In addition to the temporal and spatial stability of an environment, it is well established that life history patterns, including reproduction strategies and relationship to substrate, are of utmost importance for a species' distribution [7]. For instance, attached deep water scleractinians require consolidated substrates to survive, whilst unattached forms are found on or in unconsolidated sediments [2, 3].

Given the difficulty of sampling in deep-water marine ecosystems, the mapping and classification of biodiversity into spatial units (which then act as surrogates for unmapped biodiversity) is a common approach in spatial planning [8–12]. Considering the growing concern regarding declining ocean health, voluntary commitments to reach a national 30% area protection by 2030, and the United Nations call for better ocean governance [13–15], such spatial classifications are powerful tools to guide conservation and management strategies to support the achievement of the United Nations 14th Sustainable Development Goals (SDGs).

Despite early marine collections along South Africa's shores in the 1700's [16, 17], ocean resource management is still constrained by the poor state of knowledge of key invertebrate species, particularly offshore [18]. Endeavouring to address such species data gaps, local research advancements have recently been initiated by re-examining natural history collections [19–23]. Some of these studies have informed the national map of marine ecosystem types developed by Sink et al. [18, 24] for the National Biodiversity Assessment (NBA). The NBA used pelagic and benthic data, including biological information (macrofauna, epifauna, and fish) to produce an expert-driven ecosystem type map for national assessment and reporting. Absent, however, from this national spatial classification map is a holistic consideration of the South African azooxanthellate scleractinian fauna, as their distribution patterns had not yet been investigated. The NBA does however report on some distribution records of potential Vulnerable Marine Ecosystem indicator taxa, which includes records of two reef-building azooxanthellate coral taxa (Dendrophylliidae [25] and Caryophylliidae [26]).

Cairns [6] grouped the available literature on azooxanthellate Scleractinia into broad geographic regions, although not a biodiversity analysis, this output served as a starting point for emerging taxonomists in the field. Cairns and Keller [27] summarised depth affiliations within the southwest Indian Ocean, in which South African taxa reported off the eastern and southern margins were represented. Apart from these two publications [6, 27], the South African azooxanthellate Scleractinia distribution pattern, in relation to physical variables, has not been

investigated. This study therefore aims to understand the biodiversity gradients of the South African azooxanthellate coral fauna by analysing the environmental correlates of museum samples. To achieve this, an approach was needed to source and standardize such data from 11 surveys, including those conducted a century ago.

## 2. Material and methods

Data considered for this study were based on a subset of species distribution records for the South African azooxanthellate scleractinian fauna recently reported by Filander et al. [28]. Samples without co-ordinate were omitted from the Filander et al [28] dataset, resulting in 761 occurrence records (**Fig 1** and **S1 Appendix**: Occurrence data). These coral occurrence data were predominately collected during six historical dredge surveys undertaken between 1898 and 1990, listed below with the vessel or expedition name and depth range represented by the collection in parenthesis. These were Research Vessel (RV) *Anton Bruun* [50–1000 m], *Benguela IV* [100 m], RV *Meiring Naude* [50–1000 m], RV *Pieter Faure* [50–400 m, and 1000 m], RV *Sardinops* [50 m], and University of Cape Town Ecological Surveys [50–300 m and 500–600 m]). The recent surveys undertaken in the 21st century are represented by two trawl (*NANSEN* [50–200 m] and Department of Environment, Forestry and Fisheries/South African Environmental Observation Network demersal surveys [50–1000 m]) and three dredge surveys (ACEP: Deep–Secrets [200–500 m, 700 m and 1000 m]; IMIDA [100–200 m] surveys and Department of Environment, Forestry and Fisheries [200–500 m]).

The historical data sets had a varying degree of reliability in terms of associated data and, therefore, required data sourcing in some cases and validation in others. Consequently, all the occurrence records were first geo-referenced using ArcGIS 10.1. This step involved overlaying the coral point data on the NBA marine ecosystem types map [18, 24]. Records that were recovered on the coastline were moved to the closest polygon boundary of the ecosystem types with the near command in ArcGIS 10.1. This process was particularly beneficial for the *Pieter Faure* stations, which had positions in degrees magnetic North (not true North); whereby land bearings were used as a reference. In the next step, the spatial join tool was used to assign depth in relation to the most recent national bathymetric dataset [29] to each of the coral records, irrespective of whether depth was reported in the coral archive data set or not. The reason for this change is that modern mapping techniques have significantly improved historical bathymetric data, which were often either missing or erroneous [30]. Depth contours started at 50 m and were plotted at 100 m isobath intervals to a maximum of 1000 m, then further grouped according to shallow or deep zone. The resulting data set consisted of 95 of the total 108 azooxanthellate scleractinian species known from South Africa [28].

### 2.1. Assumptions and sampling biases

Over 80% of the resulting data are of historical origin, and therefore pose some limitations. One of these limitations is sampling coverage bias, given that past national marine surveys focused mainly on accessible nearshore areas (intertidal - ~40 m), whilst sampling in areas beyond the continental shelf [~ 50–150 m on the eastern margin, which progressively gets deeper (~ 200 m <) towards the western margin] mostly relied on international surveys (the *Pieter Faure* expeditions being an exception) [17]. These historical surveys represent decades of sampling effort but were not systematic and primarily offer presence data, with a degree of uncertainty regarding absence. The reliability of absence data in historical datasets is inherently non-linear and leads to acknowledged challenges in interpretation. This fit–for use limitation is acknowledged. However, in order to incorporate the occurrence data into the multivariate biodiversity analysis, it was necessary to make two assumptions. The first

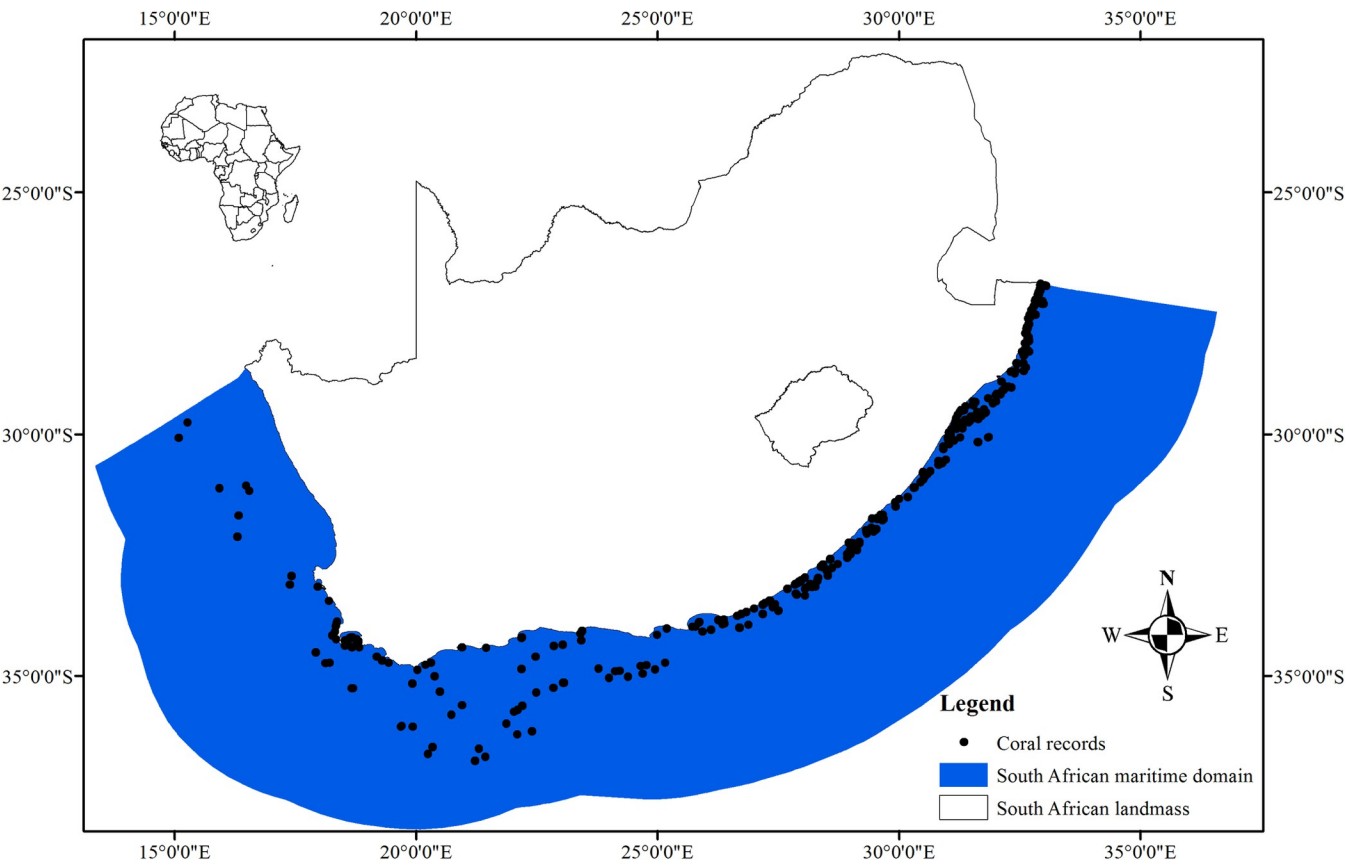

**Fig 1. Study domain and spatial coverage of the coral records forming the basis of the analysis.**

assumption involves presenting the data as presence–absence, and the second assumption involves extrapolating the occurrence of historical records to the modern day. In terms of the latter, this is a bias because anthropogenic activities (e.g., trawling) may have altered these habitats and some species may no longer be present in the historically noted area. Nonetheless, depth and co-ordinate information are the only two variables commonly associated with such datasets, noting these may be unreliable in some instances (the *Pieter Faure* collection).

Furthermore, the data preparation methodology does not follow the interpolation of the presence-absence matrix (if species occur between two extreme points, then occurrence is assumed in between) as conducted in preceding marine benthic invertebrate studies based on museum specimens [21–23]. This approach would have yielded unrealistic conclusions in the absence of fine-scale seabed data- as substrate type is one of the primary drivers of coral settlement [3]. As substratum data (grab samples and multibeam do not sufficiently cover the available coral samples [18, 31] substrate data were not considered to support interpolation techniques. The only two variables considered in this paper are longitude and depth (**S1 Appendix**: Occurrence data).

## 2.2. Data analysis

A presence-absence matrix (**S1 Appendix**: presence-absence) of the coral occurrence data was compiled and all analyses were undertaken using the PRIMER 7 software package [32], with the PERMANOVA+ add on [33]. The matrix, consisting of 488 columns (stations/

samples) and 95 rows (species), was converted to a resemblance matrix. The associated higher taxonomic classifications of these resulting species identifications were thereafter extracted from the World Register of Marine Species batch match online function [34] (**S1 Appendix**: Taxonomic attributes). Owing to the patchy nature of the data set, in which 30 species were represented by only one sample and 22 species by less than ten samples (**S1 Appendix**: Number of records per species), the Gamma+ dissimilarity matrix was selected- a measure based on average taxonomic distinctiveness. [35, 36]. This dissimilarity matrix is based on the average taxonomic distinctiveness (ATD) measure, which used the cophenetic distances derived from the phylogeny clades established in Kitahara et al. [37] and Stolarski et al. [38] (e.g. "Basal", "Complex", and "Robust") (**S1 Appendix**: Taxonomic attributes). Owing to limited resolution regarding species relationships below family level, phylogenetic scores were not assigned beyond the known molecular clades. It is important to note that ATD is a diversity calculation method that considers the distance between each species and its closest relative outside the group. The resulting ATD value provides an estimation of the group's evolutionary uniqueness, with higher values indicating greater distinctiveness. Such a procedure allowed for biotic distances among samples to be quantified even when they had zero or very few species in common [35].

The sample-specific data also required data preparation, which followed the biological data assessment. Longitude and depth are the two sample-specific variables considered to determine the environmental settings of the South African maritime domain (**S1 Appendix**: Sample-specific abiotic data). For instance if a sample was recorded at a 31˚ longitude, then it was collected in the Indian Ocean and influenced by the Agulhas Current. Each abiotic parameter was classified accordingly, before investigating the independent longitude and depth gradients (**S1 Appendix**: Sample-specific abiotic data). To classify the longitudinal data as a factor, an auto select k-R cluster mean analysis was run on the normalized longitudinal data [39]. A draftsman's plot was produced to identify the number of longitudinal groups present and validate the cluster groups present (see Fig 2). On the other hand, the depth readings were grouped according to shallow (50–200 m) *vs* deep (300–1000 m) zones.

A standard approach was undertaken to investigate changes in family attributes along the longitudinal and depth gradients, whereby a SIMPER analysis was performed to evaluate respective contributing taxa [40]. Sampling effort (denoted by N), species richness (denoted by S), Shannon index (denoted by H'log$^e$), and taxonomic distinctiveness (denoted by delta+) across the longitudinal and depth groups was quantified. The former was investigated by assigning coral records to 50x50 km grids created with the fishnet ArcGis function, whereby the grid size was guided by the boundary breaks of the k-R mean cluster groups and therefore provides information on spatial coverage of each group.

Subsequently, a RELATE routine was undertaken to evaluate if the combined longitude and depth spatial gradients correspond with those inferred from the coral species patterns [41]. Here we used the Gamma + matrix in relation to the associated depth and longitude information, which was normalised into a Euclidean distance resemblance matrix. The RELATE routine calculated a Spearman's ρ rank correlation coefficient between all elements of the coral assemblage and environmental variable resemblance matrices, followed by a permutation test. Following this, a biota and/or environment matching (BEST) test was conducted to confirm which variable contributed the most to sample statistic given by the RELATE results [42]. A species accumulation model was lastly produced to assess how well the observed azooxanthellate stony coral data represents South Africa's predicted coral diversity.

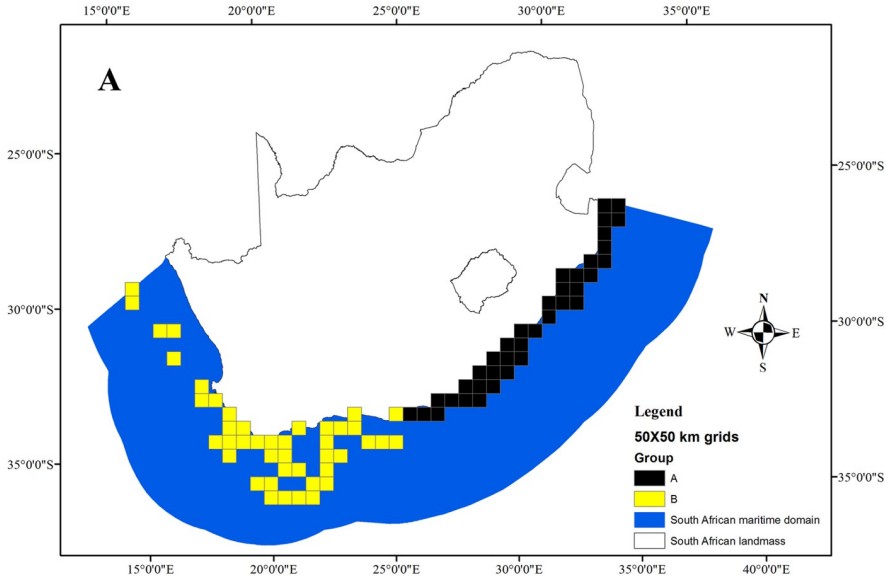

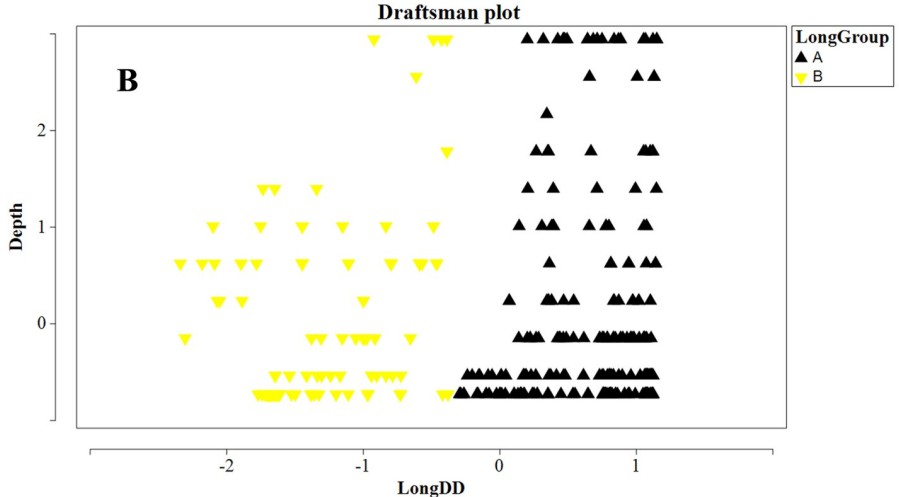

**Fig 2. A**: The 50x50 km gridded cells with samples in relation to the longitudinal groups defined by the k-R cluster analysis. Group A represents samples collected off the eastern margin and Group B are samples collected off the southern and western margins. **B**: The draftsman's plot results are also shown.

## 3. Results

### 3.1. Longitudinal gradient

The k-R (non-hierarchal) cluster analysis yielded two longitudinal groups (R = 0.94), whereby Group A encompasses samples from the eastern margin of South Africa and Group B are

**Table 1. Summary of sampling effort in relation to longitudinal gradient.**

| k-R cluster group | Number of samples | Number of 50x50 km grids | Species richness | Shannon's Index | Delta + |
|---|---|---|---|---|---|
| **A** (eastern margin) | 569 | 37 | 86 | 3.964 | 90.907 |
| **B** (southern & western margin) | 192 | 43 | 37 | 3.249 | 89.289 |

samples from the southern and western margins (**Fig 2**). The SIMPER results showed a distinction between families contributing the most to the cluster identities. Three dendrophylliids contributed the most to the similarity within group A; and the same number of caryophylliids defined Group B.

Overall, the number of samples between the two groups varied, whereby Group A (eastern margin) had more than twice the number of samples than Group B (southern and western margin) (**Table 1**). Contrary to this, the related area (number of grids) representing these samples was larger in Group B than in Group A (**Table 1**). Diversity followed the same pattern of higher measures in Group A as compared with Group B.

## 3.2. Depth gradients

A direct relationship between the number of samples (N), species richness (S), and depth was observed (**Fig 3**). The highest number of samples and observed species richness occurred between depths of 50 and 200 m, with the greatest species richness and sample count recorded at 50 m. The same two measures (S and N) fluctuated in the deep zone (300–1000 m) where the highest coral diversity measures (S and N) were recorded at 1000 m and the lowest at 800 m. Average taxonomic distinctiveness (denoted by delta +), which takes into account species phylogeny, did not show a clear pattern in coral diversity with depth and species diversity was relatively constant from 50 to 200 m. However, according to this measure, coral diversity was slightly higher at 1000 m despite the usage of a smaller number of samples from this depth (42 samples compared to 269 samples at 50 m). Eight taxonomic families were recorded at 1000 m, while only seven were recorded at 50 m. In contrast, however, the conservative Shannon diversity index mirrored the pattern of species richness with depth (**Fig 3**).

The SIMPER results of the coral species data according to family suggested that the caryophylliids, dendrophylliids, and flabellids were the main contributing taxa towards both shallow (50–200 m) and deep (300-1000m) stations. Whilst all three families collectively contributed towards the bathymetric zone comparisons (shallow *vs* deep) at a 70% cut, the Caryophylliidae representatives were more abundant at the deeper stations compared with the Dendrophylliidae and the Flabellidae in the shallow stations (**S1 Appendix**: Depth zones SIMPER results).

## 3.3. The correlation of sample-specific variables (longitude and depth groups) with coral distribution patterns

The RELATE results showed a marginal correlation (Rho-value = 0.087) but a significant difference (p-value = 0.001) when comparing the coral patterns modelled by the Gamma+ resemblance matrix with that of the Euclidean distance matrix (environmental variables - longitude and depth). It is important to note that the null hypothesis in the RELATE function is that there is no correlation. Thus, although the correlation is closer to zero (unexplained variance), the p-value confirms that longitude and depth are good predictors for the coral distribution patterns. The BEST results further confirmed the influence of depth with an independent correlation value of 0.094, whilst both environmental parameters (longitude and depth) accounted for a correlation value of 0.097.

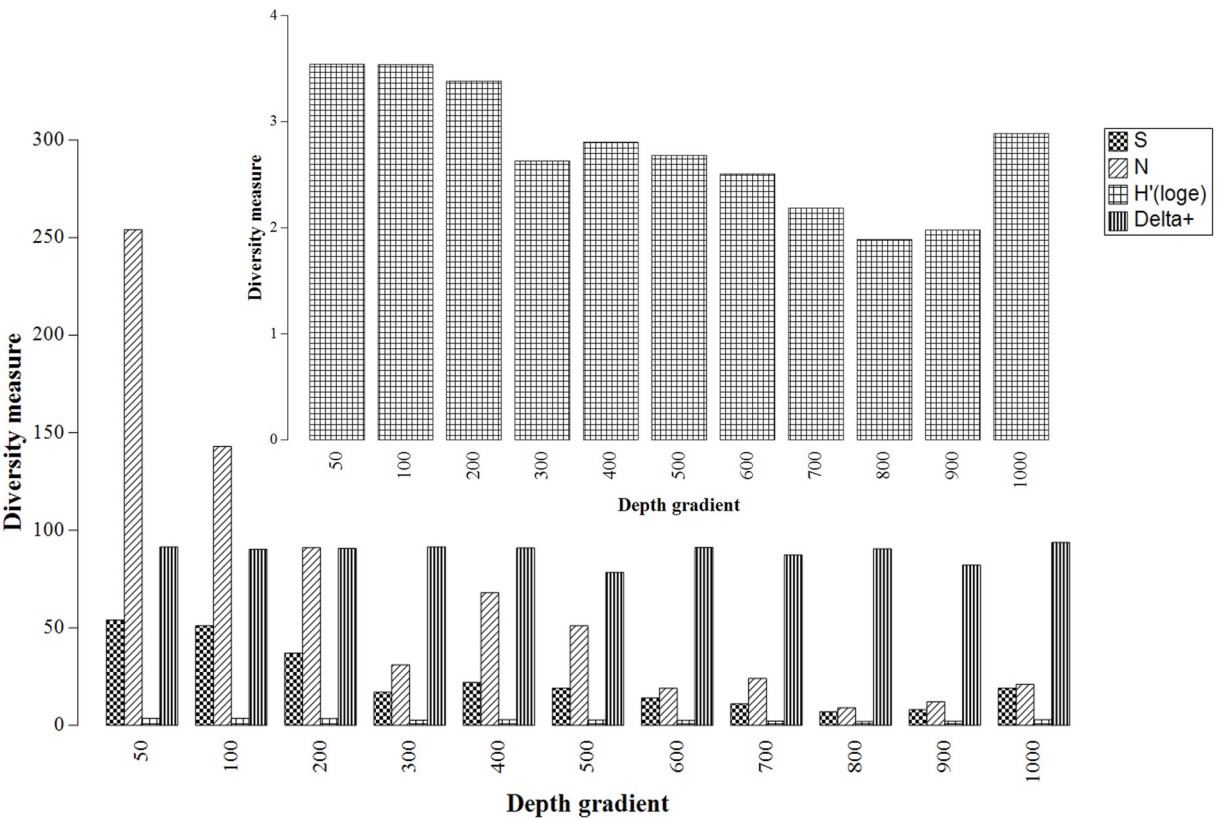

**Fig 3. The relationship between number of coral samples (N) and species richness (S) in conjunction with the average taxonomic distinctiveness (delta+) and Shannon diversity (H'log$^e$) index measures across depth gradients.** The x-axis shows samples represented in depth values in metres and the y-axis shows values that represent diversity measures in arbitrary units. The inset shows the Shannon diversity (H'log$^e$) index repeated on a Y axis of 1–4.

The majority of the species accumulation curves did not reach a plateau (**Fig 4**). All seven estimated curves, along with the observed or sampled species, started with a steep slope and indicated a rapid increase in the number of species observed with increasing sampling effort. Only two (MM and UGE) of the seven estimator curves followed the species observed pattern (Sobs), which appears to be levelling off as the sampling effort increases (**Fig 4**).

## 4. Discussion

The results of the multivariate analyses indicate that the sample-specific factors (longitude and depth) play a significant role as predictor variables for the diversity of azooxanthellate Scleractinia corals. However, there is still some unexplained variation in the data. Further investigation revealed variability among the established longitude and depth groups, and specific coral families contributing to this observed pattern were identified.

An increasing species turnover along the west to east gradient was detected. Such distributional patterns have long been reported for other South African marine invertebrates fauna (e.g., [21, 23, 43]), suggesting that different oceanographic conditions are influencing the South African marine fauna. The accompanying current regimes may also govern these contrasting species profiles across the region. Thus, although the two longitudinal boundaries (Group A = eastern margin *vs* Group B = western margin) established by the k-R mean cluster analysis do not conform to the previously proposed oceanographic boundaries [44, 45], the

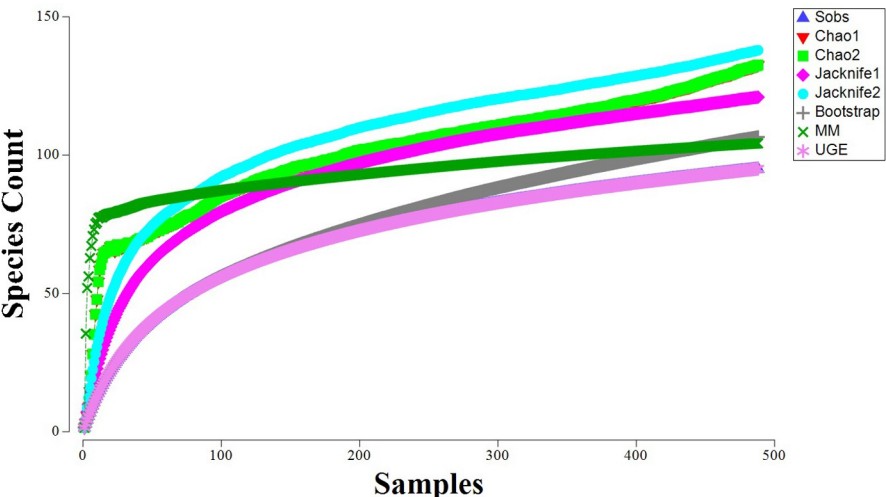

**Fig 4. Species richness accumulation curve showing the species observed (Sobs = blue upright triangle) in relation to five estimators (Chao 1 = red downward triangle, Chao 2 = green square, Jacknife 1 = pink diamond, Jacknife 2 = blue circle, Bootstrap = grey cross).** Two pairs of curves overlap, whereby the UGE estimator curve follows the same pattern as the Sobs and the Chao1 has the same pattern as Chao 2.

margins correspond to varying oceanographic variables and currents, whereby the eastern margin (Group A) is situated within the oligotrophic waters of the Indian oceanic basin and influenced by the western boundary Agulhas Current. Interestingly, Group B encompasses the southern and western margins located in both the Indian and Atlantic basins respectively. At the southern margin, the Agulhas Current retroflects, moving away from the shelf, and introduces Indo-Pacific waters into the Atlantic Ocean, the latter being regulated by the northward flowing Benguela Current [46].

The SIMPER results detailed a clear taxonomic/ family distinction within these two longitudinal groups. Dendrophylliids contributed the most to the eastern margin group whereas caryophylliids characterised the western and southern fauna. Additionally, the exclusivity in families found between Group A and Group B aligns with the proposal that species have a temperature threshold [3, 6]. The physiological characteristics of azooxanthellate coral species are indeed influenced by the properties of ambient water temperature [47, 48]. For example, an *ex-situ* experiment undertaken on the reef-building corals *D. pertusum* and *Madrepora oculata* revealed that they respond differently when exposed to three temperatures (12, 9.0, and 6.0°C; [49]). The respiration response rates varied; *M. oculata* declined whereas *D. pertusum* was not affected by temperatures being lowered. Two other physiological responses (calcification and dissolved organic carbon) were measured, and neither showed a consistent trend when comparing the two species. Thus, species belonging to different families or even congeners are expected to exhibit varying thermal tolerance.

The recovered species longitudinal pattern of low sampling effort in Group A (eastern margin) but higher number of records and diversity observed herein, was particularly surprising as the western margin (which contributes to Group B) has a higher historic sampling effort [17]. The greater presence of coral species in the eastern Agulhas region (Group A) may be explained by the heterogenous seabed provided by the increased abundance of mesophotic reefs, submarine canyons and mosaic ecosystem types [18, 24]. Whilst the incising submarine canyons along the eastern continental margin [50–53] may also give rise to a heterogenous environment, localised studies that classify different substrate types within and between

canyons are needed to confirm this hypothesis [54]. Even though the Benguela Current in the South Atlantic (influencing the western passive margin) is unique in its interactions with the western boundary Agulhas Current [44], much of this region has a unconsolidated seabed, resulting in a more homogenous environment [31, 54, 55]. Thus the unconsolidated seabed, superimposed with the slow current speed (< 3 m/s) may be a constraint for coral presence. The presence of scleractinians is however influenced by multiple factors operating at different scales, and it is crucial to consider species-specific regional adaptation abilities to environmental gradients (dissolved oxygen) - even for cosmopolitan species [56, 57]. Nonetheless, the prominence of anthropogenic activities that interact with the seabed in the Southern Benguela Upwelling area [58, 59] cannot be overlooked and may also influence the low number of species records in the area.

The southern margin is a unique area that exhibits high endemism [17]. In this region, the Agulhas Current injects Indo-Pacific waters into the Atlantic, down to depths of 2000 m in the form of anticyclonic rings [60], before retroflecting eastwards towards the Southern Indian Ocean Gyre and the Antarctic circumpolar current [61]. Schouten et al. [62] noted that the location of the retroflection loop is variable, but still within the southern region. Nonetheless, the Agulhas transport is estimated to increase from 65 Sv (1Sv - $10^6$ m$^3$s$^{-1}$) at 32˚S to 95 Sv at the southern tip of South Africa, as it breaks away from the shelf [63, 64]. Thus, the unpredictable behaviour and velocity of the Agulhas Current make this area challenging for sampling and, therefore, the low number of records here may be attributed to limited sampling effort.

The species depth gradient results complement the longitudinal species patterns whereby the univariant biodiversity measures peaked at 50 m, which corresponds to the accessible eastern margin of the South African continental maritime domain. In addition to the shelf being shallower (~ 50–150 m) and more accessible, the western boundary Agulhas Current (characteristic of this area) has been linked to the highly diverse biological properties in the Southwest Indian Ocean, where eddies can trap and transport material over long distances [65]. These complex oceanographic eddies can upwell deep nutrient-rich waters through surface divergence mechanisms [65], creating environments that favour the continuous inflow of potential coral food sources. Thus, these observations may provide grounds for a hypothesis to explain why azooxanthellate corals have a higher presence within this area. The multivariate taxonomic average distinctiveness measure (denoted by delta +) showed diversity along the depth gradients (50 to 1000 m) to be highest at 1000 m, in which eight of the 11 known South African coral families are represented. This result marginally aligns with the knowledge that the global azooxanthellate stony coral pattern has overall higher species diversity between the 200 and 1000 m [6]. Irrespective, the SIMPER analysis distinguished three major families to contribute to bathymetric zone delineation. The deeper depths (300–1000 m) were characterized by caryophylliids and flabellids, and the shallow zone (50–200 m) by dendrophylliids only. These results support the known depth affiliations of these families, in which Dendrophylliidae species occurrence is reported to peak at shallower depths (50 to 300 m) [66] and extant species of Caryophylliidae and Flabellidae are more prominent in the deeper waters (more than 200 m) [37].

The two sample-specific (depth and longitude) data sets were applied in combination to extrapolate ocean basin properties (nutrient content, salinity, temperature, etc.), which characterise the oceanographic settings influencing South African marine fauna (the colder Benguela current along the western margin, and the warmer Agulhas Current along the southern and eastern margin). In this context, the RELATE permutation model implies that longitude and depth are good predictors for coral distribution patterns. However, the close to zero R-values (R<0.5) suggests a non-linear relationship even though significant variability is evident in the species composition within the factorial groups. Whilst depth is noted to be one of the main drivers for coral distribution (as shown by BEST results), it is important to recognize that this

parameter encompasses several other properties, such as the Aragonite Saturation Horizon (ASH) that is the depth below which calcium carbonate becomes unstable and tends to dissolve [1, 67]. Such a zone has been estimated at 700–1500 m depth range south of ~ 20˚S [67]. Eight of the 11 known South African coral families are recorded within this depth range, suggesting these species are surviving within an aragonite saturation state. Interestingly, coral species have been previously reported to withstand saturating conditions [68]. The response of coral species to water properties, such as the ASH, are in no way consistent, highlighting the need for further research to comprehend the underlying environmental drivers of coral distribution.

Although the azooxanthellate coral data reported here represent an accumulation of samples over 30 years and are the best available representation of the South African fauna, all species richness estimator models did not plateau, demonstrating that the area is still not well sampled and may be much more diverse than currently known. Additional systematic sampling coverage will provide a clearer understanding of national coral diversity trends.

## 5. Conclusion and recommendations

This study examined the best available data for the South African azooxanthellate coral fauna and presented a pre-processing methodology that can provide standardised position and depth data for historical samples to allow analysis of distribution trends. Differences in azooxanthellate coral species distribution patterns across South Africa's diverse and dynamic oceanographic conditions were revealed, whereby species turnover increased on a west to east axis. A species depth gradient was additionally observed, in which the multivariate diversity measure complemented the existing knowledge on taxa trends. Despite the sparsity and unbalanced nature of the sampling, knowledge has been advanced and gaps identified. A purposeful application for this existing coral data set will be its integration into multi-taxa biogeography analyses that will support more robust data-driven ecosystem classification, description, and delineation. This in turn will support spatial prioritisation and marine spatial planning, particularly alongside taxa that share similar abiotic requirements.

## Supporting information

**S1 Appendix.**
(XLSX)

## Acknowledgments

A sincere acknowledgement goes to Dr David Herbert (Department of Natural Sciences, National Museum Wales) who assisted with associated station data for the *Meiring Naude* and *Sardinops* collections. Dr Victoria Goodall (Nelson Mandela University) for reviewing the data analysis section. Mr Ashley Johnson and Dr Lauren Williams (Department of Fisheries, Forestry, and the Environment) provided words of encouragement and ArcGIS technical support; respectively.

## Author Contributions

**Conceptualization:** Zoleka N. Filander, Kerry J. Sink, Marcelo V. Kitahara, Stephen D. Cairns, Amanda T. Lombard.

**Data curation:** Zoleka N. Filander.

**Formal analysis:** Zoleka N. Filander.

**Funding acquisition:** Zoleka N. Filander.

**Investigation:** Zoleka N. Filander.

**Methodology:** Zoleka N. Filander, Amanda T. Lombard.

**Project administration:** Zoleka N. Filander, Amanda T. Lombard.

**Resources:** Zoleka N. Filander.

**Software:** Zoleka N. Filander.

**Supervision:** Kerry J. Sink, Marcelo V. Kitahara, Stephen D. Cairns, Amanda T. Lombard.

**Validation:** Zoleka N. Filander.

**Visualization:** Zoleka N. Filander.

**Writing – original draft:** Zoleka N. Filander.

**Writing – review & editing:** Zoleka N. Filander, Kerry J. Sink, Marcelo V. Kitahara, Stephen D. Cairns, Amanda T. Lombard.

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
