## [Decision Letter · Decision Letter 0]

14 Jul 2023

PONE-D-23-17495The application of historic sample-specific variables in evaluating the biodiversity patterns of the South African azooxanthellate scleractinians (Cnidaria: Anthozoa).PLOS ONE

Dear Dr. FILANDER,

Thank you for submitting your manuscript to PLOS ONE. After careful consideration, we feel that it has merit but does not fully meet PLOS ONE’s publication criteria as it currently stands. Therefore, we invite you to submit a revised version of the manuscript that addresses the points raised during the review process. Please submit your revised manuscript by Aug 28 2023 11:59PM. If you will need more time than this to complete your revisions, please reply to this message or contact the journal office at plosone@plos.org. Please include the following items when submitting your revised manuscript:A rebuttal letter that responds to each point raised by the academic editor and reviewer(s). You should upload this letter as a separate file labeled 'Response to Reviewers'.A marked-up copy of your manuscript that highlights changes made to the original version. You should upload this as a separate file labeled 'Revised Manuscript with Track Changes'.An unmarked version of your revised paper without tracked changes. You should upload this as a separate file labeled 'Manuscript'.

We look forward to receiving your revised manuscript.

Kind regards,

Carlo Nike Bianchi

Academic Editor

PLOS ONE

Journal Requirements:

   "The financial support for this research study was provided by the Department of Forestry, Fisheries, and Environment (DFFE)."

4. We note that Figure 1 and 3 in your submission contain [map/satellite] images which may be copyrighted. All PLOS content is published under the Creative Commons Attribution License (CC BY 4.0), which means that the manuscript, images, and Supporting Information files will be freely available online, and any third party is permitted to access, download, copy, distribute, and use these materials in any way, even commercially, with proper attribution. For these reasons, we cannot publish previously copyrighted maps or satellite images created using proprietary data, such as Google software (Google Maps, Street View, and Earth). For more information, see our copyright guidelines: http://journals.plos.org/plosone/s/licenses-and-copyright.

USGS EROS (Earth Resources Observatory and Science (EROS) Center) (public domain): http://eros.usgs.gov/#+K49

Additional Editor Comments:

The ms has been red by two independent reviewers, from two countries and belonging to different academic schools. Both reviewers asked for major changes, underlining severe flaws in writing and editing. The ms needs to be largely rewritten before reconsideration by the same or other reviewers. I strongly recommend that the authors take into careful consideration all commenys of the two reviewers.

Reviewers' comments:

Reviewer's Responses to Questions

**Comments to the Author**

1. Is the manuscript technically sound, and do the data support the conclusions?

Reviewer #1: Partly

Reviewer #2: Yes

2. Has the statistical analysis been performed appropriately and rigorously? 

Reviewer #1: N/A

Reviewer #2: Yes

3. Have the authors made all data underlying the findings in their manuscript fully available?

Reviewer #1: Yes

Reviewer #2: Yes

4. Is the manuscript presented in an intelligible fashion and written in standard English?

Reviewer #1: Yes

Reviewer #2: No

5. Review Comments to the Author

Reviewer #1: In general the ms should be rearranged in material and methods and it should be cleared some points and doubts that I have outlined in the notes inside the .pdf.

Some points are not clear in the methods and the methods should be well understood and it should be replicable.

I suggest the ms should not be accepted in the present form.

Reviewer #2: I started reading this manuscript in detail and making notes where the English was awkward and needed editing, but soon ran out of time and energy to do this. The data are quite thin, although are worthy of presentation, but i feel that the existing MS is way too long and complicated to present what is available. It also needs severe editing, not only to make it shorter and simpler but also to correct a whole lot of minor editorial inconsistencies. For example to consistently capitalize proper names such as Agulhas Current, to consistently give numbers over 10 as numerals not words and the names of taxonomic families consistently in either lower case English form or capitalized Latin format (preferred).

The reference list is also a mess, firstly i do not understand why references are numbered when they are referred to by name and not number in the text . Secondly there is every type of formatting error here, from a mixture of abbreviated and non- abbreviated journal titles, to missing publication information, to spelling, spacing and punctuation errors! In short this all needs to be done a lot more carefully!

As regards the actual factual contend i think this does have merit and that a shorter, simpler and better edited version would be worthy of publication. I would particularly like to see though, some better analysis of the relationships between habitat area, number of samples taken and numbers of taxa found - for example it appears that huge swathes of deeper water on the east coast are not or hardly sampled - it would be nice to know how big these areas are and how many samples (even if zero) are available from each depth zone.This would better inform whether there are indeed few spcies or just few or no samples here.

By the way i thought the figures were overly garishly coloured!

6. PLOS authors have the option to publish the peer review history of their article (what does this mean?). If published, this will include your full peer review and any attached files.

Reviewer #1: **Yes: **Andrea Peirano

Reviewer #2: No

---

## [Author Response · Author response to Decision Letter 0]

29 Sep 2023

Nelson Mandela University

University Way

Summerstrand

Gqeberha

6019

South Africa 

15 September 2023

PLOS ONE 

1265 Battery Street, Suite 200 

San Francisco, 

CA 94111

United states of America 

Dear Dr Carlo Nike Bianchi (Academic Editor)

Re: Rebuttal Letter for Manuscript ID [PONE-D-23-17495] 

Title of Your Manuscript: The application of historic sample-specific variables in evaluating the biodiversity patterns of the South African azooxanthellate scleractinians (Cnidaria: Anthozoa).

I hope this letter finds you well. I would like to express my gratitude for the careful consideration and feedback provided by the reviewers on the above-mentioned manuscript submitted to PLOS ONE. I appreciate the time and effort the reviewers and the editorial team have invested in evaluating my work and their constructive suggestions.

I have thoroughly considered the reviewers' comments and suggestions, and I would like to demonstrate how we have addressed their concerns and provide clarifications on the points raised. Below, I have outlined each major concern and the corresponding response.

Regards, Zoleka Filander (on behalf of co-authors) 

Reviewer 1 

Reviewer #1 overview: In general the ms should be rearranged in material and methods and it should be cleared some points and doubts that I have outlined in the notes inside the .pdf.

Some points are not clear in the methods and the methods should be well understood and it should be replicable. I suggest the ms should not be accepted in the present form.

Authors response: We thank the reviewer for the suggestion and have rearranged the manuscript, provided clarity on the methods, and made changes with a view to ensuring that the methods are repeatable by other studies.

Response to comments in pdf

Reviewer #1 comment: Line 37 (now line 44 in updated manuscript) = I suggest to change into a more apropriate 

deep water coral. You examine depth between 50 and 700 m.

Authors response: “stony Cold water” in the keywords has been updated to deep water as suggested.

Reviewer #1 comment: Line 66-71 (now 95-100 in updated manuscript) = I do not agree with these sentence. biological pattern in deep area have been described in quite all the oceans around the word. Perhaps you mean in South Africa ? I suggest to change coral species instead of species. 

Authors response: We thank the reviewer for this feedback and have updated this sentence to reflect the reference to coral species and have also updated citations. 

Reviewer #1 comment: Line 90-91 (now line 133-135 in updated manuscript) = I do not understand: a taxonomic revision doesn't mean that prevoius works cannot be reconsidered and corrected.

Authors response: We thank the reviewer for highlighting this, and have updated the sentence to provide improved clarity.

Reviewer #1 comment: Line 103-105 (now line 146-150 in updated manuscript) = this last sentence should be more specific on the author's study and its objectives

you may consider some sentences in the in material and method and/or abstract

Authors response: We thank the reviewer for this suggestion and have added a summary of the overall objectives at the end of this paragraph.

Reviewer #1 comment: Line 112 (now line 179-186 in updated manuscript) = please add the range of sample depth for each cited cruise

Authors response: We thank the reviewer for this suggestion and depth ranges represented by the samples from each cruise have now been added as a table.

Reviewer #1 comment: Line 127 (now line 193 in updated manuscript) = move the references to the end of the sentence.

Authors response: The citation has been added to the end of the sentence.

Reviewer #1 comment: Line 126-127 (now line 190 in updated manuscript) = please describe the GIS system. For example what is a polygon boundary ? why you decide to '..move the sampling points to the closest polygon ? You can add a paragraph to describe the GIS architecture.

Authors response: We thank the reviewer for bringing ambiguity to our attention, and have therefore refined the opening of the sentence to provide rationale for undertaking the ArcGis steps. 

Reviewer #1 comment: Line 127-128 (now line 195-197 in updated manuscript) = this should be moved in results

Authors response: This is statement was not a result reached through this study, but rather an explanation/supporting statement as to why some of the records fell on 

land/coastline and is therefore not moved or changed. 

Reviewer #1 comment: Line 131-132 (now line 225-227 in updated manuscript) = this sentence sounds very strange .Some lines above you write you move to a polygon, (I think taking no account of the depth sampling) Now you say that recorded sampling depth were not taken in account.

These considerations should be well argued because the value of a sample (also from a hystorical point of view) is based on the data recorded (Where, When and How it was sampled) mainly vessel position and depth. From your sentences this data seems to have no value. 

Authors response: We thank the reviewer for this feedback and would like to clarify that the reasons to standardize the depths associated with each record is not to discount historical datasets but rather to establish a uniform methodology across the dataset – especially considering that such datasets are known to have various degrees of reliability. 

Furthermore, in the assumption and sampling biases sections, specifically line 233-247 of updated manuscript, we are talking about the sampling footprint in South Africa, which is higher in the nearshore areas vs the offshore areas. Historically, the latter relied on international surveys and samples are mostly hosted at international institutions (e.g., Natural history museum in London). Please take note that this paragraph (line 233-247 now in updated manuscript) is different to the previous paragraph (line 190-230 in updated manuscript) - which is explaining why the two-part ArcGis approach was undertaken. We have clarified this by adding a sentence outlining the improvement of current bathymetry data vs historical techniques

Reviewer #1 comment: Line 139 (now 235-237 in updated manuscript) = depth range ?

Authors response: We thank the reviewer for the suggestion and have updated by adding the depth ranges.

Reviewer #1 comment: Line 142 (now line 238-239 in updated manuscript) = why they are biased

Authors response: We assume the data to be presence-absence in the multivariate analysis, when in actual fact the data are presence-only. However, collections, and the resolution of collected data, depend on the expertise onboard on any given expedition. In other words, just because a coral specimen was not recorded/preserved, it doesn’t mean that it was not present at a sampled station. We have therefore left this statement as is and no changes have been made. 

Reviewer #1 comment: Line 146-147 (now line 244-247 in updated manuscript) = this could be an error. Hystorical data can give informations on the appearance/disappearance of species in time. For example, the hystorical occurence/not occurrence of a species is now used as a method to evaluate cliatic changes in biological communities.

Authors response: We thank the reviewers for this comment and have updated to clarify the reasoning. As additional information- we are assuming that the records are where they were historically recorded, and this might not be the case considering the footprint of anthropogenic activities. Whilst historical datasets can be used to investigate changes in biological communities, the anthropogenic footprint needs to be considered. This has been difficult to isolate.

Reviewer #1 comment: Line 153-154 (now line 261-264 in updated manuscript) = this interpolation is not a general assupmtion. It depends on the geographical scale.

Authors response: We agree, it does depend on scale, but this approach was not considered due to the uneven availability of substrate information in South Africa - as highlighted in lines 266-271. 

Reviewer #1 comment: Line 155-157 (now line 264-271in updated manuscript) = In this part is introduced the bottom characteristcs and discussed, it is the first time that bottom characteristic is cited

I suggest to indroduce at the start of the material and methods all parameters collected in the hystorical paper research or published literature, than discuss each one ( depth, species, latitude/longitude, type of bottom etc) and why and how was included in your dataset

Authors response: in the Introduction (now lines 78-82) in the updated manuscript) we mention the importance of habitat/ substrate/sediment type to the survival of coral. These sentences (lines 264-271) are included to support why the interpolation technique was not applied. This has been clarified by adding a sentence that highlights that the only two variables used in this paper are longitude and depth (bearing in mind that this section is outlining the assumptions made and the biases of the dataset, before unpacking the multivariate analysis conducted and the subsequent results).

Reviewer #1 comment: Line 161 (now line 271 in updated manuscript) = what is a substrate level ? Do you mean substrate type ? Or level is a matrix level or GIS data level ?

Authors response: We have clarified that substrate refers to characteristics of the seabed (i.e., sediment/substrate data). The updated paragraph (264-272) reflects the uneven representation of substrate information in South Africa. 

Reviewer #1 comment: Line 163 = please describe ATD

Reviewer #1 comment: Line 164-174 = this part is a conclusion ?

Authors response: We have clarified that Average Taxonomic distinctiveness is a diversity measure that considered phylogenetic data, and added sentences on how this is quantified, along with reasoning for using ATD. 

This paragraph has been deleted in the “assumption and biases: section and important information on ATD incorporated into the first paragraph of the data analysis section (now line 340-351 in updated manuscript). 

Reviewer #1 comment: Line 210-21 (now line 367-368 in updated manuscript) = this approach should be introduced early, where you describe the database and/or the GIS approach.

Authors response: We thank the reviewer for this suggestion and have refined the existing sentence in line 228-229 (in updated manuscript), whilst also keeping it in this section (line 367-368 in updated manuscript) for ease of reading, we see this section to be also appropriate in the data analysis.

Reviewer #1 comment: Line 380 (now line 762 in updated manuscript) = Madrepora

Authors response: We thank the reviewer for bringing out attention to this typo, which has now been updated.

Reviewer #1 comment: Line 438 (now 836-837 in updated manuscript) = in the material and methods you write there are two great dividion 50-200 and 200-1000 with steps of 100 m.

Hence, the eigth families are between 900 and 1000 m ?

Authors response: We thank the reviewer for the comment and have updated the sentences to reflect that the diversity gradients were quantified based on the associated sample depth data. Thus eight families are represented at 1000 m - this statement is independent of the zone groupings (i.e., 50-200 m vs 300-1000 m) used/needed in the SIMPER analysis to evaluate the characteristic species of the zones which is discussed in the following sentences (now line 840-849 in updated manuscript).

Reviewer #1 comment: Line 443-445 (now line 843-849 in updated manuscript) = so, if i have understood you find that dendrophyllidae fhave a range limited to a maximum of 200 m if compared with Cairns (2001)

Authors response: Our findings support Cairn’s (2001) statement that dendrophylliids are most abundant in the 50-300 m depth. We have clarified this in the manuscript by amending the sentence to reflect this. 

Reviewer 2 

Reviewer #2 comment: I started reading this manuscript in detail and making notes where the English was awkward and needed editing, but soon ran out of time and energy to do this. The data are quite thin, although are worthy of presentation, but i feel that the existing MS is way too long and complicated to present what is available. It also needs severe editing, not only to make it shorter and simpler but also to correct a whole lot of minor editorial inconsistencies. For example to consistently capitalize proper names such as Agulhas Current, to consistently give numbers over 10 as numerals not words and the names of taxonomic families consistently in either lower case English form or capitalized Latin format (preferred).

Authors response: We have severely edited this manuscript and thank the reviewer for the recommendations to simplify and improve the manuscript. The manuscript has been shortened and simplified by –

1. Removing the ANOSIM methodology and results, thus focusing on the gradients of each variable (longitude and depth) and the corelation between the biological vs environmental (i.e., longitude and depth) data. 

2. Removing the paragraph on the species characteristic of the factors being tested (longitude and depth) and focusing on the family distinction only. 

3. A substantial edit to reduce unnecessary text and shorten the manuscript as far as possible.

Proper nouns have been updated and are capitalized, and the number over 10 has been updated to a numerical.

In terms of the taxonomic families, the English and Latin format has been maintained, as this provides concise reading. Using both has been done in other published work. See - 

https://doi.org/10.1016/S1055-7903(03)00162-3

https://doi.org/10.1016/j.ympev.2022.107565

https://doi.org/10.1038/s41598-020-77763-y

Reviewer #2 comment: The reference list is also a mess, firstly i do not understand why references are numbered when they are referred to by name and not number in the text . 

Authors response: Thank you for the comment, we have maintained the numbering as the journal guidelines require references to be numbered 

Secondly there is every type of formatting error here, from a mixture of abbreviated and non- abbreviated journal titles, to missing publication information, to spelling, spacing and punctuation errors! In short this all needs to be done a lot more carefully!

Authors response: We apologise for this oversight and have updated each reference with increased attention to detail.

Reviewer #2 comment: As regards the actual factual contend i think this does have merit and that a shorter, simpler and better edited version would be worthy of publication. I would particularly like to see though, some better analysis of the relationships between habitat area, number of samples taken and numbers of taxa found - for example it appears that huge swathes of deeper water on the east coast are not or hardly sampled - it would be nice to know how big these areas are and how many samples (even if zero) are available from each depth zone.This would better inform whether there are indeed few spcies or just few or no samples here.

Authors response: We sincerely appreciate the reviewer’s comment and would like to bring to their attention that detailed information regarding the geographical representation of each longitudinal group can be found in Table 1, which is included within the text. Additionally, the depth bar graph (Fig. 4) illustrates the distribution of records/species across the different isobaths. 

We would like to clarify that the data presented in our study is presence only data, but for the purposes of the multivariate analysis, it is considered as presence-absence. Consequently any absence of of data should not be interpreted as a true representation of the dataset. We kindly request that an absence analysis of the dataset is not pursued. 

Reviewer #2 comment: By the way i thought the figures were overly garishly coloured!

Authors response: We thank the reviewer for his comment and have changed the colour to monochrome where possible.

---

## [Decision Letter · Decision Letter 1]

15 Nov 2023

PONE-D-23-17495R1The application of historic sample-specific variables in evaluating the biodiversity patterns of the South African azooxanthellate scleractinians (Cnidaria: Anthozoa).PLOS ONE

Dear Dr. FILANDER,

Thank you for submitting your manuscript to PLOS ONE. After careful consideration, we feel that it has merit but does not fully meet PLOS ONE’s publication criteria as it currently stands. Therefore, we invite you to submit a revised version of the manuscript that addresses the points raised during the review process.

 Please submit your revised manuscript by Dec 30 2023 11:59PM. If you will need more time than this to complete your revisions, please reply to this message or contact the journal office at plosone@plos.org. Please include the following items when submitting your revised manuscript:A rebuttal letter that responds to each point raised by the academic editor and reviewer(s). You should upload this letter as a separate file labeled 'Response to Reviewers'.A marked-up copy of your manuscript that highlights changes made to the original version. You should upload this as a separate file labeled 'Revised Manuscript with Track Changes'.An unmarked version of your revised paper without tracked changes. You should upload this as a separate file labeled 'Manuscript'.If applicable, we recommend that you deposit your laboratory protocols in protocols.io to enhance the reproducibility of your results. Protocols.io assigns your protocol its own identifier (DOI) so that it can be cited independently in the future. For instructions see: https://journals.plos.org/plosone/s/submission-guidelines#loc-laboratory-protocols. Additionally, PLOS ONE offers an option for publishing peer-reviewed Lab Protocol articles, which describe protocols hosted on protocols.io. Read more information on sharing protocols at https://plos.org/protocols?utm_medium=editorial-email&utm_source=authorletters&utm_campaign=protocols.

We look forward to receiving your revised manuscript.

Kind regards,

Carlo Nike Bianchi

Academic Editor

PLOS ONE

Journal Requirements:

**Additional Editor Comments:**

One of the former reviewers, while appreciating that the ms has been improved, thinks that some work of revision is still necessary. Apart from the specific points indicated, the ms is considered lengthy. The Authors are therefore required to be more synthetic.

Reviewers' comments:

Reviewer's Responses to Questions

**Comments to the Author**

1. If the authors have adequately addressed your comments raised in a previous round of review and you feel that this manuscript is now acceptable for publication, you may indicate that here to bypass the “Comments to the Author” section, enter your conflict of interest statement in the “Confidential to Editor” section, and submit your "Accept" recommendation.

Reviewer #2: (No Response)

2. Is the manuscript technically sound, and do the data support the conclusions?

Reviewer #2: Yes

3. Has the statistical analysis been performed appropriately and rigorously? 

Reviewer #2: Yes

4. Have the authors made all data underlying the findings in their manuscript fully available?

Reviewer #2: Yes

5. Is the manuscript presented in an intelligible fashion and written in standard English?

Reviewer #2: Yes

6. Review Comments to the Author

Reviewer #2: a large number of revisions and corrections have been made to this version which now reads much more clearly (although still long in relation to the data content in my opinion!

I spotted only a small number of points that still need correction as follows ( indicated by line number):

83 - i presume you mean 'since the 1700's'

98- it is not necessary of useful to insert 'i.e. ' when detailing a set of items in brackets. This happens many times in MS and should be removed

101-103 i suggest merging these two sentences instead or repeating the reference

149 - data are pleural ( of datum), so 'are' not 'is'

288- space needed before m

Refs 8-9 words in the one title given with caps, in the other in lower case - which is correct format?

ref 27 - in different font to others

587 - there is a spare M in this line

596 full stop needed after title

623 Seconded? do you mean Second Edition?

643 i think title is Research not Res, which is abbreviation

644 no journal given for this ref

Ref 92 why are words capitalised in this title

Figs 1 and 2 could easily be combined by adding blacks to the first figure

7. PLOS authors have the option to publish the peer review history of their article (what does this mean?). If published, this will include your full peer review and any attached files.

Reviewer #2: **Yes: **Charles Griffiths

---

## [Author Response · Author response to Decision Letter 1]

4 Dec 2023

Nelson Mandela University

University Way

Summerstrand

Gqeberha

6019

South Africa 

01 December 2023

PLOS ONE 

1265 Battery Street, Suite 200 

San Francisco, 

CA 94111

United states of America 

Dear Dr Carlo Nike Bianchi (Academic Editor)

Re: Rebuttal Letter for Manuscript ID [PONE-D-23-17495] 

Initial title of manuscript: The application of historic sample-specific variables in evaluating the biodiversity patterns of the South African azooxanthellate scleractinians (Cnidaria: Anthozoa).

Revised title of manuscript: Diversity patterns of the South African azooxanthellate scleractinians (Cnidaria: Anthozoa), with considerations of environmental correlates.

I trust this correspondence finds you in good health. I want to convey my appreciation for the meticulous evaluation and feedback given by the reviewer(s) regarding the submitted manuscript to PLOS ONE. The dedication of both the reviewer and the editorial team in assessing this work and providing constructive suggestions is genuinely valued.

We have carefully examined the reviewers' comments and suggestions, and I wish to illustrate our responsiveness to their concerns. Below, we have delineated each major concern along with its corresponding response.

Regards, 

Zoleka Filander (on behalf of co-authors) 

Reviewer 2 

Reviewer #2 overview: a large number of revisions and corrections have been made to this version which now reads much more clearly (although still long in relation to the data content in my opinion!

Authors response: We thank the reviewer for the suggestion to compress the contents within the manuscript. This has prompted improved referencing of the methodology and the elimination of redundant/repetitive information. The manuscript has undergone rearrangement, resulting in a reduction of the text from 33 to 31 pages. To enhance representation and reference convenience, Figure 2 has been integrated with the draftsman's plot results, and the permutation histogram of the RELATE results has been omitted. These comprehensive edits have necessitated a title change, aligning more accurately with the manuscript's refined content.

Reviewer #2 comment: I spotted only a small number of points that still need correction as follows ( indicated by line number):

• 83 - i presume you mean 'since the 1700's'

• 98- it is not necessary of useful to insert 'i.e. ' when detailing a set of items in brackets. This happens many times in MS and should be removed

• 101-103 i suggest merging these two sentences instead or repeating the reference

• 149 - data are pleural ( of datum), so 'are' not 'is'

• 288- space needed before m

• Refs 8-9 words in the one title given with caps, in the other in lower case - which is correct format?

• ref 27 - in different font to others

• 587 - there is a spare M in this line

• 596 full stop needed after title

• 623 Seconded? do you mean Second Edition?

• 643 i think title is Research not Res, which is abbreviation

• 644 no journal given for this ref

• Ref 92 why are words capitalised in this title

• Figs 1 and 2 could easily be combined by adding blacks to the first figure

Authors response: All the aforementioned suggestions have been incorporated, with revisions made to rectify these errors, and the manuscript has undergone significant editing..

---

## [Editor Report · Decision Letter 2]

8 Dec 2023

Diversity patterns of the South African azooxanthellate scleractinians (Cnidaria: Anthozoa), with considerations of environmental correlates.

PONE-D-23-17495R2

Dear Dr. FILANDER,

We’re pleased to inform you that your manuscript has been judged scientifically suitable for publication and will be formally accepted for publication once it meets all outstanding technical requirements.

Kind regards,

Carlo Nike Bianchi

Academic Editor

PLOS ONE
---

## [Editor Report · Acceptance letter]

2 Jul 2024

PONE-D-23-17495R2 

PLOS ONE

Dear Dr. Filander, 

I'm pleased to inform you that your manuscript has been deemed suitable for publication in PLOS ONE. Congratulations! Your manuscript is now being handed over to our production team.

Kind regards, 

on behalf of

Dr. Carlo Nike Bianchi 

Academic Editor

PLOS ONE